## Research Article

food insecurity; micronutrient deficiency; dietary diversity; adolescents; depression; anxiety

**Corresponding author:**
Zulfiqar Bhutta;
Email: zulfiqar.bhutta@aku.edu

Susan Campisi and Florence Perquier indicates equal contribution.

# Impact of household food insecurity and nutrition on depression and anxiety symptoms among adolescents living in rural Pakistan

Susan Campisi[1,2,3,4] ⓘ, Florence Perquier[1] ⓘ, Yaqub Wasan[5] ⓘ, Sajid Soofi[5] ⓘ, Daphne Korczak[2,6] ⓘ, Suneeta Monga[2,6] ⓘ, Peter Szatmari[1,2,6] and Zulfiqar Bhutta[4,6] ⓘ

[1]Cundill Centre for Child and Youth Depression, Centre for Addiction and Mental Health, Toronto, ON, Canada; [2]Department of Psychiatry, Hospital for Sick Children, Toronto, ON, Canada; [3]Nutrition and Dietetics Program, Clinical Public Health Division, Dalla Lana School of Public Health, University of Toronto, ON, Canada; [4]Centre for Global Child Health, Hospital for Sick Children, Peter Gilgan Centre for Research and Learning, Toronto, ON, Canada; [5]Centre of Excellence in Women and Child Health, Aga Khan University, Karachi, Pakistan and [6]Department of Psychiatry, Temerty Faculty of Medicine, University of Toronto, Toronto, ON, Canada

## Abstract

**Background:** This study investigates the links between dietary diversity, food insecurity and mental health (depression and anxiety) in adolescents from rural Pakistan. Adolescence is a critical time for developing mental health disorders, yet limited research exists on these issues in low- and middle-income countries (LMICs).

**Methods:** The study included 1,396 adolescents (ages 9–15) and assessed their mental health, nutrition and maternal well-being. Depression and anxiety were measured using standardized questionnaires, while dietary diversity and food insecurity were evaluated through household assessments. Incidence rate ratios assessed the relationship between nutrition and mental health.

**Results:** Results showed that 8.1% of boys and 10.2% of girls experienced depression, with anxiety rates ranging from 5.8% to 39.1%. Adolescents from households with higher dietary diversity had lower symptoms of depression and anxiety (IRRs:0.91–0.96), while those with higher food insecurity had increased symptoms (IRRs:1.24–1.86). Folate deficiency was associated with depressive symptoms, particularly in boys. Maternal mental health was observed to mediate the relationship between food insecurity and adolescent depression and anxiety.

**Conclusions:** The study highlights that improving maternal mental health and addressing nutritional deficiencies, particularly folate, may benefit adolescent well-being. Further research in other LMICs is needed to explore these associations and their mechanisms.

## Impact statement

This study offers crucial insights into the impact of nutrition, food insecurity and maternal mental health on adolescent well-being in rural Pakistan, a region where mental health issues are often overlooked. With a large number of adolescents living in low- and middle-income countries (LMICs) like Pakistan, this research is particularly important as it addresses an underexplored area in global health. By demonstrating the significant role of household dietary diversity in reducing symptoms of depression and anxiety, and revealing the mediating effect of maternal mental health, the study highlights key pathways for improving adolescent mental health in these regions.

The findings underscore the importance of addressing adolescent nutritional deficiencies, particularly folate and household food insecurity to support better mental health outcomes. The study also emphasizes the need to consider maternal mental health as a key factor in adolescent well-being, providing a broader understanding of the social determinants affecting mental health. These insights are critical for designing interventions aimed at reducing mental health disparities in LMICs.

This research has the potential to inform public health policies and programs that promote access to diverse, nutritious diets at the household level and improve maternal mental health, which could significantly reduce the burden of mental health disorders in adolescents. It also lays the groundwork for future studies in other LMICs, encouraging further exploration of nutrition and mental health interventions as effective strategies for improving adolescent well-being. By highlighting the intersection of nutrition, food security and mental health, this study contributes to a more holistic approach to supporting vulnerable populations worldwide.

## Introduction

During adolescence, a substantial proportion of youth experience diagnosable mental disorders. Epidemiological research indicates that approximately 13% of adolescents have a diagnosed condition, with anxiety and depressive disorders being particularly prevalent, accounting for an estimated 40% (UNICEF, 2017). This is consistent with the broader global trend of deteriorating adolescent mental health since 2010, exacerbated by the COVID-19 pandemic (McGorry et al., 2024). This trend is concerning, especially considering the large number of adolescents living in South Asia. Home to 340 million adolescents, South Asia has the highest concentration globally of adolescents (UNICEF, 2017). Consequently, South Asia also has the highest burden of mental health issues in this age group, with an estimated 42 million adolescents suffering from a mental disorder (Keeley, 2021).

In Pakistan, depression is the greatest contributor to disability-adjusted life years (DALYs), followed by anxiety disorders across the life span (Alvi et al., 2023). Although national depression and anxiety rates specific to adolescents in Pakistan are not readily available, various studies point towards an upward trend. The prevalence of anxiety and depression in Pakistan has steadily risen, increasing from 9% and 11% between 1992 and 2009 to 17% and 21% in 2018, respectively(Khalid et al., 2019; Sarwat et al., 2009). This mirrors the global trajectory of increasing mental health challenges among adolescents (Piao et al., 2022). Furthermore, the comorbidity of depression and anxiety disorders in children and adolescents can range from 16% to 62% (Al-Asadi et al., 2015). The repercussions of anxiety and depressive disorders extend beyond adolescence. Studies have demonstrated links to cardiometabolic conditions such as diabetes and cardiovascular disease in adults (Scott, 2014; Shrestha and Copenhaver, 2015). Additionally, adolescent depression and anxiety can negatively impact other areas of life, being associated with risky behaviour (Pozuelo et al., 2022), poor school performance, (Khesht-Masjedi et al., 2019; Osborn et al., 2020) and death by suicide (Grossberg and Rice, 2023). Thus, developing effective and universally accessible interventions for depression and anxiety in adolescents is critical.

First-line adolescent mental healthcare treatment, such as medications and psychotherapy, is scarce in many low- and middle-income countries (LMIC) such as Pakistan. Barriers to accessing these treatments, which include stigma, low literacy rates, inadequate data collection and a shortage of clinicians, can further hinder the effectiveness of mental healthcare for children and adolescents in these regions (Chachar and Mian, 2022). Even when accessible, first-line treatments may not be universally effective, with up to 40% of depressed adolescents not responding to evidence-based treatment (Dwyer et al., 2020). Given these challenges, innovative, universally accessible interventions are crucial. Emerging research suggests lifestyle interventions, such as improvements in dietary quality, hold significant potential. (Marx et al., 2023)

A growing body of research demonstrates a significant association between poor dietary intake and depression in adolescents (Korczak et al., 2021; Mooreville et al., 2014; Orlando et al., 2021) as well as anxiety (Collins et al., 2021; Gibson-Smith et al., 2018; Jacka et al., 2011). Unhealthy eating behaviours likely become more prevalent during adolescence due to increased peer influence and a shift towards more meals consumed outside the family home. Numerous studies conducted in North America and Europe have consistently demonstrated that adolescents in these regions consume insufficient amounts of fruits and vegetables while overindulging in unhealthy

dietary components, such as total fat, saturated fats, sugars and sodium (Flieh et al., 2021; Golper et al., 2021; Hack et al., 2021). Pakistani adolescents, mirroring their global counterparts, exhibit a common tendency to deviate from Pakistan's recommended dietary guidelines (Amjad et al., 2022; Food and Agriculture Organization of the United Nations and Government of Pakistan, 2018; Gill et al., 2016; Munir et al., 2023). This trend may contribute to the increased vulnerability to mental health issues observed in this age group, highlighting the potential role of dietary quality in adolescent mental well-being.

Adolescents in LMIC are more likely to face food insecurity, which can compound their poor food choices. Research has shown a clear association between greater food insecurity and increased depression and anxiety among adolescents (Smith et al., 2023). Pakistan, like many developing countries, faces ongoing challenges related to food security. Despite improvements in national food security between 2012 and 2022, the 2022 Global Food Security Index categorizes Pakistan as a high food insecurity risk country (Economist Intelligence Unit, 2022). This concern is further underscored by the 2018 National Nutrition Survey, which reveals that 37% of the population faces food insecurity (Government of Pakistan, 2019). While data specifically on adolescent food insecurity in Pakistan is limited, the national risk classification suggests a potential vulnerability for this age group. Consistent with the national risk classification, a study published before the COVID-19 pandemic reported that adolescent food insecurity was found among 52% of adolescents living in rural Pakistan (Sheikh et al., 2020). This is comparable to other developing countries such as Ethiopia and Lebanon, where 50% of adolescents are food insecure (Belachew et al., 2012; Jomaa et al., 2019). Moreover, food-insecure adolescents continue to favour unhealthy food choices. A study among 68 countries including Pakistan conducted using the Global School-based Student Health Survey, found that food insecurity was associated with 1.17 times higher odds (95% CI 1.08, 1.26) of fast-food consumption among adolescents aged 12–15 years (Smith et al., 2022).

This study aims to investigate the associations between variations in household dietary diversity, household food insecurity, depression and anxiety symptoms in adolescents residing in rural Pakistan. Elucidating these relationships may inform the development of culturally appropriate and resource-sensitive interventions to address adolescent mental health challenges.

## Methods

### Participants

The Nash-wo-Numa study is a cross-sectional study conducted between January 2019 and February 2020 in the rural district of Matiari, located in the Sindh province of Pakistan. Boys and girls between 9 and 15.9 years of age and their birth mothers were assessed for inclusion. Data from a previous census study, conducted among 53,000 households, allowed for the identification of eligible households (Pakistan Bureau of Statistics, 2017). A computer-assisted random sampling procedure was then used to select households, depending on the sex and age of children, and the number of occupants in the household (Campisi et al., 2019). Detailed methods have been published elsewhere (Campisi et al., 2019). Staff members visited each household. Out of 1,873 eligible households, 212 households were excluded from the study sample because of age criteria (*n* = 151); because the mother was not available, did not consent to participate or could not be interviewed

for cognitive reasons ($n = 9$); because adolescents suffered from a chronic or genetic illness that might have impacted their development or because girls were or had been pregnant ($n = 52$). For the present study, we also excluded participants with no mental health data. The final sample consisted of 1,396 participants (678 boys and 718 girls).

This study received ethics approvals from the Aga Khan University, Karachi, Pakistan Ethics Review Committee and SickKids Hospital, Toronto, Canada Research Ethics Board, and the Centre for Addiction and Mental Health Research Ethics Committee. Informed written consent from parents or legal guardians and written assent from adolescents were obtained.

## Procedure

After giving their consent to participate in the study, adolescents and their mothers were invited to visit the nearest field-based office. A chaperone could accompany them if necessary. Staff members arranged the appointment and transportation. At the field-based office, data were collected during face-to-face interviews with trained staff members, using questionnaires translated into Sindhi (the language spoken by the majority of the population of Matiari). All the questionnaires were translated and reviewed by the team members and psychologists, who spoke both English and Sindhi and tested in a pilot study. Trained psychologists administered mental health measures at field-based offices. Questionnaires were administered to adolescents in the presence of their mothers.

Aga Khan University (AKU) has conducted several studies within the District of Matiari. During the pilot testing of data collection tools, the community was consulted to ensure the use of understandable local terms, particularly regarding the presence of mothers during child psychological assessments and maintaining privacy during pubertal assessments. The study team also conducted village-level meetings with community gatekeepers and villagers to inform them about the study's aims and procedures. Such meetings are typically held by AKU's teams before the start of any study to prevent misconceptions and promote clear understanding within the community.

## Mental health measures

*The Short Mood and Feelings Questionnaire (SMFQ)* was used to assess depression symptoms experienced by adolescents (Angold et al., 1995; Rhew et al., 2010). Adolescents were asked to rate how they had been feeling or acting recently on 13 items. Each item was rated using a 3-point Likert Scale ("not true" = 0, "sometimes" = 1 or "true" = 2) and the total score ranged from 0 to 26. In the Nashwo-Numa study, the SMFQ demonstrated strong unidimensionality and good internal consistency (Shetty et al., 2022). A cut-off point of SMFQ > 8 indicated significant depressive symptoms. A higher score indicates more depressive symptoms.

*The Screen for Child Anxiety-related Disorders (SCARED)* was administered to adolescents and their mothers to assess symptoms of anxiety experienced by adolescents over the past three months (Beidas et al., 2015; Birmaher et al., 1997). The SCARED was used to measure total or specific anxiety using five subscales: panic disorder or significant somatic symptoms, generalized anxiety disorder, separation anxiety disorder, social anxiety disorder and school avoidance. For each subscale, thresholds have been suggested to indicate the presence of clinically significant anxiety disorders: panic/somatic disorder (score $\geq 7$), generalized anxiety disorder ($\geq 9$), separation anxiety disorder ($\geq 5$), social anxiety disorder

($\geq 8$) and school avoidance ($\geq 3$) (Birmaher et al., 1997). Nair et al. used a similar approach to describe anxiety symptoms experienced by adolescents living in a rural community in India (Nair et al., 2013). In our sample, only 74.9% of boys and 52.2% of girls in our sample attended school. The main reason for not attending school was a "lack of interest" (reported by 52.9% of boys and 36.7% of girls). The four SCARED items on school avoidance were poorly reported by adolescents who did not attend school and were excluded from the computation of the total SCARED score, in order to ensure a reliable comparison between adolescents who attended and did not attend school. The total SCARED score was then computed using the 37 items of the four remaining scales, each being rated as 0 (Not true or hardly ever true), 1 (Sometimes true) or 2 (True or often true). In our study, the internal consistency of this revised SCARED scale was excellent ($\alpha = 0.93$). The total score ranges from 0 to 74. A higher score indicates greater anxiety symptoms.

*The Warwick-Edinburgh Mental Wellbeing Scale (WEMWBS)* was administered to mothers to rate their general mental well-being (Stewart-Brown and Janmohamed, 2008). Each of the 14 items was rated from 1 (none of the time) to 5 (all of the time), with higher scores indicating higher levels of mental well-being. The scale has been validated in numerous settings including Pakistan (Waqas et al., 2015). The total score ranges from 14 (lowest) to 70 (highest). A higher score indicates better mental well-being.

## Nutrition measures

*Household Dietary Diversity Score (HDDS)* is a validated tool used to estimate household food group consumption and serves as a proxy measure of household food access (Kennedy et al., 2011). HDDS has been widely used to assess the household dietary diversity of adolescents and measures the number of food groups consumed by a household within a specific timeframe (Gyimah et al., 2021; Islam et al., 2020). Studies have shown that a more diversified household diet is associated with higher caloric and protein intake, a greater percentage of protein from animal sources and higher household income (Swindale and Bilinsky, 2006). Mothers were asked for information on the household consumption of 17 food groupings (foods made from grains; white roots, tubers and plantains; pulses; nuts and seeds; dark green leafy vegetables; vitamin-A rich vegetables; other vegetables; vitamin-A rich fruits; other fruits; organ meat; meat and poultry; eggs; fish and seafood; milk and milk products; oils and fats; sweets and sugar; condiments) in the preceding 7 days (Kennedy et al., 2011; Swindale and Bilinsky, 2006). Each food group was assigned a score of 1 if consumed and 0 if not. The household's overall HDDS score ranged from 0 to 17, with higher scores indicating a more diverse diet.

*Household Food Insecurity Experience Scale (FIES)* was used to assess food-insecurity experiences of children and adolescents within households over the last 12 months (Ballard et al., 2013). Since 2014, the FIES had been included in the Gallup World Poll to obtain cross-culturally comparable estimates of food insecurity in more than 150 countries internationally (Saint Ville et al., 2019). The FIES examines food insecurity as a "lived experience" and has been widely used among adolescents (Frongillo, 2013; Fusta et al., 2023; Pereira et al., 2021; Valente et al., 2024). Mothers were asked to report whether they had experienced any of eight situations of food insecurity (i.e. worrying about not having enough food to eat because of a lack of money or other resources). Each item was coded as 1 (yes) or 0 (no). The FIES score was obtained by summing up the

score of each item and dichotomized as food secure (0–3) and moderately or severely food insecure (4–8).

*Body mass index (BMI) and Stunting.* The trained study staff at the field-based clinic measured participants' height and weight. All measurements were conducted at least twice using a Seca digital floor scale (model 813) and a Seca stadiometer (model 213), with participants wearing light clothing and being barefoot (Campisi et al., 2019). Height and weight were estimated to the nearest 0.1 kilogram and centimetre. A third measurement was taken if the first two differed by more than 1 cm in height and more than 0.5 kg in weight. The mean of the measures was used to compute BMI *z*-score according to the World Health Organization (WHO) age- and gender-specific growth standards (de Onis M et al., 2007). BMI-*z* scores were classified as severe thinness (< −3SD), thinness (−3 ≤ BMI z-score <−2), normal (−2SD ≤ BMI *z*-score < 1SD) and overweight/obesity (>1SD) (de Onis M et al., 2007). Height for age *z*-scores (HAZ) was also calculated according to WHO growth standards and stunting was defined as HAZ < −2SD. A combined stunting and thinness variable was defined as HAZ < −2 and BMI *z*-score ≤ −2.

*Micronutrient serum levels and anemia.* When adolescents agreed to a blood draw, and the parent gave their consent, a total of 5 mL whole blood was collected by venipuncture into zinc-free tubes. Serum concentrations of vitamin A (retinol), vitamin D (25 (OH)D), folate, zinc and ferritin were treated as continuous variables, while the deficiency status was categorized based on specific cutoff values (Institute of Medicine Committee to Review Dietary Reference Intakes for Vitamin and Calcium, 2011; King et al., 2015; Maier et al., 2006; WHO, 2011a, b, 2012, 2020). The variable "multiple micronutrient deficiencies" was defined as having deficiencies in three or more of these nutrients: vitamin A, vitamin D, folate, zinc and ferritin.

Haemoglobin (Hb) testing was performed by drawing a drop of blood from adolescents' fingertips. Mild anemia was defined as a Hb concentration between 11 and 11.49 g/dL in adolescents younger than 12 years old, between 11 and 11.99 g/dL in adolescents aged 12–14.9, and from 11 to 12.99 g/dl in boys aged 15 or older) (World Health Organization, 2011a). Moderate to severe anemia was defined as a Hb concentration lower than 11.0 g/dL.

*Covariates.* Covariates known to influence adolescent mental health or nutritional status were included in our analyses. These included the child's age (as a continuous variable), the child's school attendance (yes/no) and whether mothers had an occupation (yes/no).

### Statistical analysis

Continuous variables were described using means and standard deviations and categorical variables were presented as numbers and percentages. Analyses were stratified by sex as age ranges differed between boys and girls.

To assess the relationship between nutrition measures and mental health outcomes, the association of nutrition measures with depression and anxiety symptom scores was examined using incident rate ratios (IRR) estimated through negative binomial regression models. Negative binomial regression was selected over Poisson regression due to the highly positively skewed distribution of the MFQ and SCARED scores and significant overdispersion. Each nutrition variable was first included in a model adjusted for age, school attendance and mother's occupation (Model 1). Variables significantly associated with depression or anxiety symptom scores were then introduced in a multi-adjusted model, which

was also adjusted for the presence of any other deficiency (Model 2). Mother's mental well-being was included in a third model (Model 3).

Structural equation models were used to better understand the role of mother's mental well-being and household dietary diversity in the relationship between household food insecurity and adolescent depression and anxiety symptoms (Hatem et al., 2020). In our models, household food insecurity was suggested to influence mental health symptoms either directly or indirectly, through household dietary diversity or the mother's mental well-being (Miller et al., 2021). Models were adjusted for age, school attendance, mother's occupation and BMI. To explore the potential role of financial support in the relationship between nutrition measures and anxiety symptoms in boys, sensitivity analyses were conducted for Models 3 and 4. Adjusting for financial support did not alter the results (data not shown). All analyses were performed using STATA version 16 (StataCorp LP, College Station, Texas) and a *p*-value < .05 was considered statistically significant.

### Results

The mean age of participants was 12.9 ± 1.6 years in boys and 12.0 ± 1.7 years in girls. The prevalence of depression was 8.1% in boys and 10.2% in girls. The prevalence of anxiety disorders ranged from 5.8% for generalized anxiety disorder in boys to 39.1% for separation anxiety disorder in girls. Almost 25% of households experienced moderate or severe levels of food insecurity and 57.1% of households were below the official government extreme poverty threshold and received financial support in the form of cash, food vouchers, a scholarship or through the Benazir Income Support Program. Statistically significant differences between boys and girls were observed across BMI parameters, and micronutrient serum concentrations and deficiencies. See Table 1 for all participant, maternal and household characteristics.

Both male and female households demonstrated limited dietary diversity among nutritive foods. Milk, milk products, grains and non-nutritive foods (sweets, sugar, oils, fats, other beverages, other foods and condiments) were consumed weekly (Figure 1). White roots, tubers, plantains, vitamin A-rich vegetables and other vegetables were part of the household diet four times a week. In contrast, dark green leafy vegetables and pulses (beans, peas, lentils) were consumed two to three times a week; Vitamin A-rich fruits, other fruits, meat, poultry and eggs one to two times a week; and fish, other seafood, organ meat, nuts and seeds were consumed less than once a week. (Supplementary Table 1). Food group consumption was similar among male and female households except for vitamin A-rich vegetables eaten more often in male households (*p* = .031). HDDS was highly correlated (*r* > 0.50) with the consumption of fruit (Vitamin A containing fruit and other fruit), animal protein (organ meat, eggs and meat protein) and fish (Supplementary Table 2). FIES was correlated with HDDS but not with food group consumption or serum micronutrient concentrations (Supplementary Table 2).

The results of the multivariate models are summarized in Table 2. Model 1, adjusted for age, and school attendance, revealed significant associations between nutrition measures and mental health symptoms in both boys and girls. Specifically, dietary diversity, food insecurity and folate deficiency were significantly associated with greater depression symptoms in both sexes (respectively, for boys, IRR = 0.906; CI 95% = 0.856–0.958; *p* = .002, IRR = 1.863; CI95% = 1.474–2.354; *p* < .001, IRR = 1.771; CI95% = 1.261–2.487;

**Table 1.** participant characteristics

| | Boys | | Girls | | |
| --- | --- | --- | --- | --- | --- |
| | n | % or mean [SD] | n | % or mean [SD] | *p*-value |
| **Participant characteristics** | | | | | |
| Age in years, mean[SD] | 678 | 12.9 [1.6] | 718 | 12.0 [1.7] | **<.001** |
| School attendance | 508 | 74.9 | 375 | 52.2 | **<.001** |
| **Anxiety** | | | | | |
| Anxiety symptom score (SCARED)[a] | 678 | 12.5 [9.9] | 718 | 16.7 [11.8] | **<.001** |
| **Anxiety subscale** | | | | | |
| Panic disorder or significant somatic disorder (subscale score ≥ 7) | 678 | 13.3 | 718 | 17.1 | **0.045** |
| Generalized anxiety disorder (subscale score ≥ 9) | 678 | 5.8 | 718 | 7.7 | 0.155 |
| Separation anxiety disorder (subscale score ≥ 5) | 678 | 23.9 | 718 | 39.1 | **<.001** |
| Social anxiety disorder (subscale score ≥ 8) | 678 | 11.2 | 718 | 22.3 | **<.001** |
| School avoidance (among those attending school) (subscale score ≥ 3) | 508 | 15.8 | 375 | 15.5 | 0.909 |
| **Depression** | | | | | |
| Depression symptom score (SMFQ)[b] | 678 | 2.3 [3.2] | 718 | 2.7 [3.5] | **0.004** |
| Prevalence of depression (SMFQ > 8) | | 8.1 | | 10.2 | 0.184 |
| **Body mass index (BMI) *z*-score** | 673 | −1.5 [1.2] | 718 | −1.2 [1.1] | **<.001** |
| **BMI *z*-score categories** | | | | | |
| Severe thinness (BMI *z*-score < −3 SD) | 67 | 9.9 | 35 | 4.9 | **<.001** |
| Thinness (−3 ≤ BMI *z*-score < −2) | 147 | 21.7 | 116 | 16.2 | |
| Normal | 439 | 64.8 | 545 | 75.9 | |
| Overweight/obese (BMI z-score >2 SD) | 20 | 3.0 | 22 | 3.1 | |
| Missing | 5 | 0.7 | 0 | 0.0 | |
| **Stunting (HAZ < −2)** | 186 | 27.4 | 273 | 38.0 | **<.001** |
| **Stunting and thinness** | 82 | 12.1 | 77 | 10.7 | 0.421 |
| **Serum levels** | | | | | |
| Hemoglobin concentration (g/dL) | 673 | 12.8 [1.5] | 718 | 12.1 [1.6] | **<.001** |
| Non anemic | 529 | 78.6 | 480 | 66.9 | **<.001** |
| Mild anemia | 79 | 11.7 | 99 | 13.8 | |
| Moderate anemia | 56 | 8.3 | 125 | 17.4 | |
| Severe anemia | 9 | 1.3 | 14 | 2.0 | |
| Serum ferritin concentration (ng/mL) | 673 | 41.9 [72.2] | 717 | 31.9 [28.7] | **<.001** |
| Normal ferritin | 501 | 74.4 | 467 | 65.1 | |
| Iron deficient anemia (IDA)[c] | 172 | 25.6 | 250 | 34.9 | |
| Serum 5(OH)D concentration (ng/mL) | 671 | 18.0 [6.9] | 714 | 13.1 [6.8] | **<.001** |
| Vitamin D deficiency (<12 ng/mL) | 140 | 20.9 | 362 | 50.7 | **<.001** |
| Vitamin A/serum retinol concentration (μg/dL) | 672 | 24.1 [12.1] | 708 | 19.9 [10.2] | **<.001** |
| Normal Vitamin A (>70 μmol/l) | 384 | 57.1 | 283 | 40.0 | **<.001** |
| Mild deficiency (0.36–0.70 μmol/l) | 229 | 34.1 | 331 | 46.8 | |
| Severe deficiency (≤ 0.35 μmol/l) | 59 | 8.8 | 94 | 13.3 | |
| Serum zinc concentration (μg/dL) | 670 | 62.4 [18.0] | 715 | 64.7 [21.2] | 0.050 |
| Zinc – normal[h] | 161 | 24.0 | 292 | 40.8 | **<.001** |
| Zinc deficiency[g] | 509 | 76.0 | 423 | 59.2 | |

(Continued)

**Table 1.** (*Continued*)

| | Boys | | Girls | | |
|---|---|---|---|---|---|
| | n | % or mean [SD] | n | % or mean [SD] | *p*-value |
| **Participant characteristics** | | | | | |
| Serum folate concentration (ng/mL) | 589 | 4.4 [3.8] | 587 | 4.9 [3.6] | **<.001** |
| Serum folate levels | | | | | |
| Normal (6−20 ng/L) | 100 | 14.8 | 127 | 17.7 | **<.001** |
| Possible deficiency (3−5.9 ng/L) | 236 | 34.8 | 273 | 38.0 | |
| Deficient (<3 ng/L) | 253 | 37.3 | 187 | 26.0 | |
| Missing | 89 | 13.1 | 131 | 18.3 | |
| Multiple micronutrient deficiencies (≥3)[i] | 203 | 29.9 | 252 | 35.1 | **<.001** |
| **Maternal characteristics** | | | | | |
| Mother's working status | | | | | |
| Homemaker | 404 | 59.6 | 445 | 62.0 | 0.360 |
| Working | 274 | 40.4 | 273 | 38.0 | |
| Mother's mental health well-being score[e] | 678 | 51.9 [9.7] | 718 | 51.8 [10.1] | |
| **Household characteristics** | | | | | |
| Household financial support[d] | 404 | 59.6 | 393 | 54.7 | 0.067 |
| Household dietary diversity score (HDDs)[f] | 678 | 12.6 [2.0] | 718 | 12.5 [1.9] | 0.686 |
| Food Insecurity Experience Scale (FIES)[g] | 678 | 1.6 [2.8] | 718 | 1.5 [2.7] | 0.809 |
| Prevalence of moderate or severe food insecurity (FIES ≥ 4) | 165 | 24.3 | 167 | 23.3 | 0.637 |

*Note*: Bold values indicate statistical significance.
[a]Score on the Short Mood and Feelings Questionnaire (SMFQ)
[b]Score on the Screen for Child Anxiety-related Disorders (SCARED).
[c]<15 ng/mL if no inflammation/ <70 ng/mL if inflammation.
[d]Cash, Food voucher, scholarship or Benazir Income support Program.
[e]Score on the Warwick-Edinburgh Mental Wellbeing Scale (WEMWBS).
[f]Score on the Household Dietary Diversity Score (HDDs).
[g]Score on the Household Food Insecurity Experience Scale (FIES).
[h]For children under 10 years of age, a morning non-fasting plasma zinc concentration less than 65µg/dL for girls and an afternoon plasma zinc concentration less than 57 µg/dL was classified as zinc deficiency; girls over 10 years of age with a morning non-fasting zinc concentration of less than 66 µg/dL or an afternoon zinc concentration of less than 59 µg/dl; in boys over 10, and afternoon zinc concentration of 61 µg/dL.
[i]Based on serum concentration of vitamin A, Vitamin D, Folate, Zinc, and Ferritin. Possible deficiency was not included in the definition of folate deficiency. 231 missing data (92 in boys, 139 in girls).

*p* = .001; for girls, IRR-0.919; CI95% = 0.873–0.968; *p* = .001, IRR = 1.584; CI95% = 1.276–1.967; *p* < .001, IRR = 1.631; CI95% = 1.211–2.197; *p* = .001) and higher anxiety symptom scores (respectively, for boys, IRR = 0.960; CI 95% = 0.932–0.099; *p* = .009, IRR = 1.424; CI95% = 1.244–1.632; *p* < .001, IRR = 1.272; CI95% = 1.051–1.539; *p* = .013; for girls, IRR = 0.944; CI95% = 0.918–0.971; *p* < .001, IRR = 1.370; CI95% = 1.216–1.543; *p* < .001, IRR = 1.292; CI95% = 1.107–1.508; *p* = .001). Ferritin deficiency was also significantly associated with higher anxiety scores in boys (IRR = 1.154; CI95% = 1.006–1.324; *p* = .041). While thinness (IRR = 0.738; CI95% = 0.565–0.963; *p* = .025) and obesity (IRR = 0.496; CI95% = 0.25 = 0.982; *p* = .044) were significantly associated with lower symptoms of depression but not anxiety in boys; obesity was significantly associated with lower symptoms of depression (IRR = 0.673; CI95% = 0.467–0.968; *p* = .033) in girls.

Model 2 was adjusted for age, school attendance, mother's occupation and the presence of any other deficiency. In boys, food insecurity was associated with higher depression and anxiety symptoms (respectively, IRR = 1.52; CI95% = 1.17–1.99; *p* = .002 and IRR = 1.33; CI95% = 1.14–1.55; *p*<.001). Higher depression symptoms were also found in boys with probable folate deficiency (IRR = 1.43; CI95% = 1.02–2.00; *p* = 0.036) and actual folate deficiency

(IRR = 1.53; CI95% = 1.08–2.16; *p* = 0.016). In girls, a higher dietary diversity score was associated with lower anxiety symptoms (IRR = 0.96; CI95% = 0.94–0.99; *p* = 0.014) while food insecurity was associated with increased anxiety symptoms (IRR = 1.22; CI95% = 1.07–1.39; *p* = 0.003). Food insecurity was potentially associated with increased depression symptoms (IRR = 1.26; CI 95% = 1.00–1.60; *p* = 0.052). Higher levels of depression symptoms were observed in girls with possible and actual folate deficiency (respectively IRR = 1.42; CI95% = 1.08–1.87; *p* = .012 and IRR = 1.53; CI95% = 1.14–2.05; *p* = .005), whereas increased anxiety symptoms were found in girls with actual folate deficiency only (IRR = 1.22; CI 95% = 1.04–1.43; *p* = .014).

After the inclusion of the mother's mental health in the models, only the associations observed with folate serum levels remained significant (Table 2, Model 3). Food insecurity was potentially associated with increased anxiety symptoms in boys (IRR = 1.15; CI95% = 1.00–1.33; *p* = .050) and depressive symptoms in girls (IRR = 2.58; CI95% = 0.99–6.71; *p* = .051). A significant interaction between food insecurity and the mother's mental health was found in relation to depression symptoms (*p* = .042) in girls and was included in the corresponding model. Probable and actual folate deficiency was associated with higher depression scores in boys

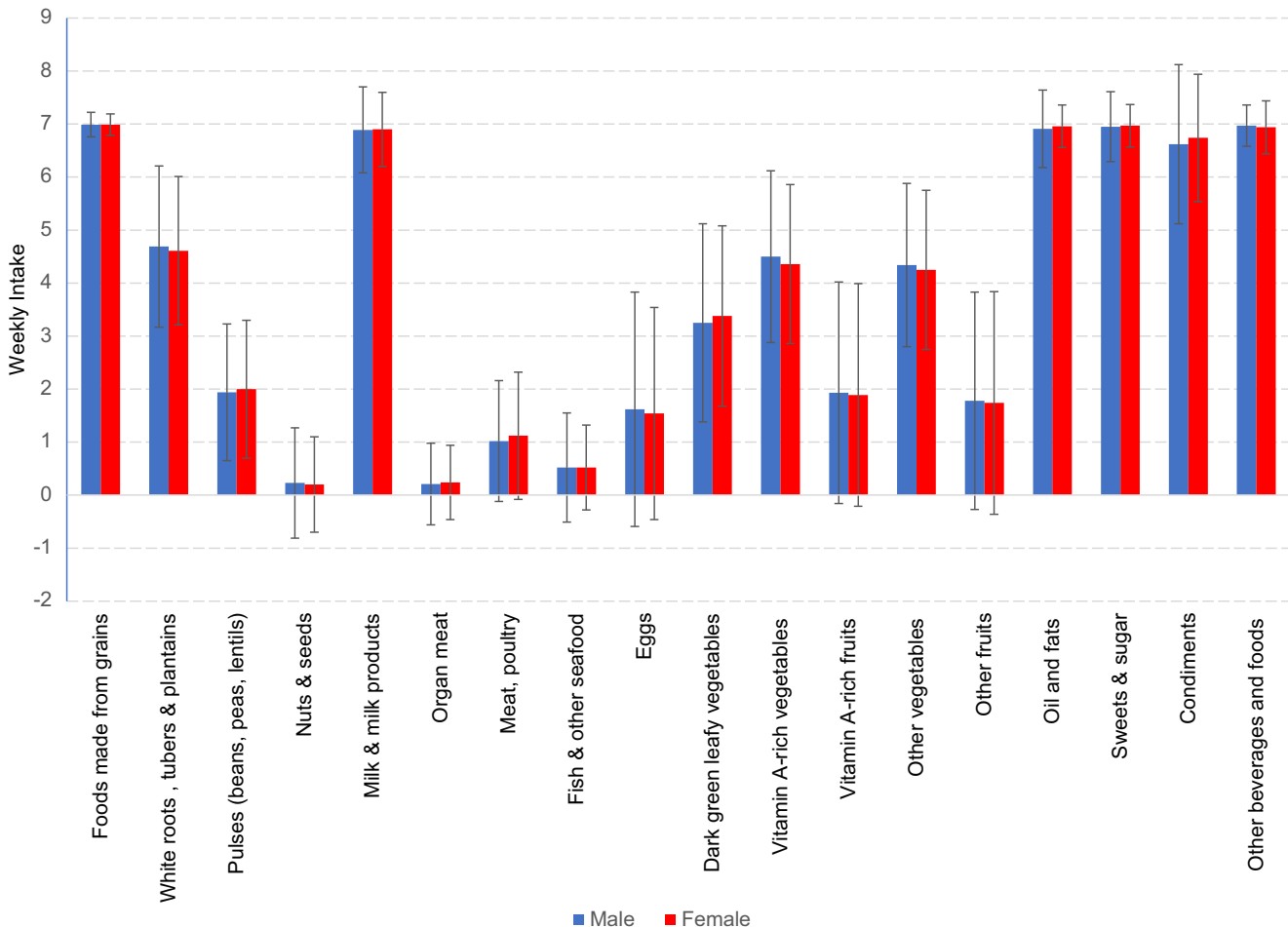

**Figure 1.** Frequency of household food group consumption, by sex.

(respectively IRR = 1.38; CI95% = 1.01–1.89; $p$ = .045 and IRR = 1.47; CI95% = 1.06–2.03; $p$ = .019; and girls (respectively IRR = 1.31; CI95% = 1.02–1.69; $p$ = .036 and IRR = 1.32; CI95% = 1.00–1.73; $p$ = .050). Girls with folate deficiency also presented a higher risk of anxiety symptoms (IRR = 1.16; CI95% = 1.00–1.35; $p$ = .046).

Structural equation models allowed for a more nuanced understanding of the association between mother's mental health, food insecurity and mental health symptoms (Figure 2). Food insecurity was negatively associated with dietary diversity and mother's mental well-being in boys (respectively, $\beta$ = −.284, $p$ < .001 and $\beta$ = −.840, $p$ < .001) and girls (respectively, $\beta$ = −.194, $p$ < .001 and $\beta$ = −.704, $p$ < .001). Mother's mental well-being was found to be associated with both food insecurity and adolescents' mental health, suggesting a potential indirect relationship between the two. Higher scores on the mother's mental well-being measure were significantly associated with a decrease in depression and anxiety symptoms in boys (respectively, $\beta$ = −.130, $p$ < .001 and $\beta$ = −.395, $p$ < .001) and in girls (respectively, $\beta$ = −.143, $p$ < .001 and $\beta$ = −.468, $p$ < .001). No significant associations were found between dietary diversity and adolescent's mental health. A significant positive relationship between food insecurity and increased symptoms of depression ($\beta$ = 0.115, $p$ = 0.011) and anxiety ($\beta$ = 0.514, $p$ < .0001) was found in boys, but not in girls. In girls only, the mother's mental well-being was positively associated with dietary diversity ($\beta$ = .020, $p$ = .003).

## Discussion

This study offers insight into the relationship between nutritional measures and mental health symptoms among adolescents in rural LMIC settings, expanding on the limited research in this area. The high levels of nutritional biomarker deficiencies combined with low BMI observed in this group indicate widespread malnutrition, likely driven by limited dietary diversity and food insecurity. The findings reveal a significant association between nutritional factors – especially food insecurity and folate deficiency – and greater symptoms of depression and anxiety observed among adolescents living in Matiari. Despite regular consumption of staple foods like grains and dairy, the infrequent intake of other vital food groups, such as meat, poultry, eggs, fruits and vegetables, points to an urgent need for greater dietary diversity. Furthermore, the study highlights the pivotal role of maternal mental health in mediating the effects of food insecurity on adolescent mental health, underscoring the importance of a holistic approach to addressing these issues.

This study adds robust evidence to the growing body of research linking nutrition with adolescent mental health and extends it to LMIC. The findings reveal significant associations among household dietary diversity, household food insecurity experiences and folate deficiency with more symptoms of depression and anxiety in both boys and girls. These results are consistent with previous

**Table 2.** Univariate and multivariate associations of nutrition related variables with depression and anxiety symptoms, by sex

| Model | | | | | | | | | | | | | | | | | |
|---|---|---|---|---|---|---|---|---|---|---|---|---|---|---|---|---|---|
| | Boys (n = 678) | | | | | | | | Girls (n = 718) | | | | | | | | |
| | Depressive symptoms (SMFQ) | | | | Anxiety symptoms (SCARED) | | | | Depressive symptoms (SMFQ) | | | | Anxiety symptoms (SCARED) | | | |
| Model 1: adjusted for age and school attendance | n | IRR | [95%CI] | P>z | IRR | [95%CI] | P>z | n | IRR | [95%CI] | P>z | IRR | [95%CI] | P>z |
| Food diversity and insecurity | | | | | | | | | | | | | | | | | |
| HDDS score | 678 | 0.906 | 0.856,0.958 | **.001** | 0.960 | 0.932,0.99 | **.009** | 718 | 0.919 | 0.873,0.968 | **.001** | 0.944 | 0.918,0.971 | **<.001** |
| Moderate/severe FIES | 678 | 1.863 | 1.474,2.354 | **<.001** | 1.424 | 1.244,1.632 | **<.001** | 718 | 1.584 | 1.276,1.967 | **<.001** | 1.370 | 1.216,1.543 | **<.001** |
| Serum concentrations (continuous) | | | | | | | | | | | | | | | | | |
| Vitamin A | 672 | 1.001 | 0.992,1.01 | .854 | 1.000 | 0.995,1.005 | .859 | 708 | 0.998 | 0.989,1.007 | .659 | 0.999 | 0.994,1.004 | .627 |
| Vitamin D | 671 | 0.985 | 0.97,1.001 | .072 | 0.992 | 0.984,1.001 | .095 | 714 | 0.996 | 0.982,1.011 | .626 | 0.998 | 0.99,1.006 | .546 |
| Folate | 589 | 0.967 | 0.938,0.996 | **.025** | 0.990 | 0.973,1.007 | .240 | 587 | 0.976 | 0.95,1.004 | .096 | 0.983 | 0.969,0.998 | **.028** |
| Zinc | 670 | 0.999 | 0.993,1.005 | .775 | 1.000 | 0.996,1.003 | .844 | 715 | 0.998 | 0.993,1.002 | .350 | 0.999 | 0.996,1.001 | .296 |
| Ferritin | 673 | 1.000 | 0.999,1.002 | .605 | 1.000 | 0.999,1.000 | .365 | 717 | 1.000 | 0.997,1.003 | .944 | 1.000 | 0.998,1.002 | .896 |
| Serum concentrations (deficiencies) | | | | | | | | | | | | | | | | | |
| Vitamin A | 672 | | | | | | | 708 | | | | | | |
| Mild deficiency | | 0.916 | 0.727,1.156 | .460 | 0.982 | 0.862,1.119 | .784 | | 1.100 | 0.897,1.348 | .361 | 1.037 | 0.928,1.159 | .521 |
| Severe deficiency | | 1.091 | 0.739,1.61 | .662 | 1.006 | 0.805,1.258 | .957 | | 0.98 | 0.725,1.325 | .895 | 1.071 | 0.911,1.26 | .406 |
| Vitamin D deficiency | 671 | | | | | | | 714 | | | | | | |
| Insufficiency | | 1.137 | 0.89,1.452 | .305 | 1.064 | 0.928,1.22 | .375 | | 0.925 | 0.684,1.252 | .614 | 1.012 | 0.859,1.193 | .885 |
| Deficiency | | 1.188 | 0.879,1.605 | .263 | 1.040 | 0.878,1.232 | .651 | | 1.044 | 0.777,1.404 | .775 | 1.048 | 0.892,1.232 | .565 |
| Folate | 589 | | | | | | | 587 | | | | | | |
| Possible deficiency | | 1.390 | 0.993,1.946 | .055 | 1.030 | 0.853,1.244 | .757 | | 1.432 | 1.083,1.894 | **.012** | 1.155 | 1,1.334 | .050 |
| Deficiency | | 1.771 | 1.261,2.487 | **.001** | 1.272 | 1.051,1.539 | **.013** | | 1.631 | 1.211,2.197 | **.001** | 1.292 | 1.107,1.508 | **.001** |
| Zinc deficiency | 670 | 0.987 | 0.768,1.269 | .921 | 0.897 | 0.78,1.032 | .129 | 715 | 1.027 | 0.849,1.244 | .782 | 1.032 | 0.929,1.145 | .559 |
| Ferritin deficiency | 673 | 0.994 | 0.776,1.273 | .964 | 1.154 | 1.006,1.324 | **.041** | 717 | 1.048 | 0.86,1.278 | .639 | 0.979 | 0.878,1.091 | .701 |
| Anemia | 673 | | | | | | | 718 | | | | | | |
| Mild | | 0.968 | 0.69,1.357 | .851 | 0.973 | 0.802,1.18 | .778 | | 0.845 | 0.638,1.119 | .241 | 0.950 | 0.816,1.106 | .509 |
| Moderate/severe | | 0.834 | 0.577,1.206 | .334 | 1.053 | 0.858,1.292 | .623 | | 0.909 | 0.713,1.158 | .439 | 0.937 | 0.821,1.07 | .339 |

(*Continued*)

**Table 2.** (*Continued*)

| Model | Boys (*n* = 678) | | | | | | Girls (*n* = 718) | | | | | | |
|---|---|---|---|---|---|---|---|---|---|---|---|---|---|
| | | Depressive symptoms (SMFQ) | | | Anxiety symptoms (SCARED) | | | | Depressive symptoms (SMFQ) | | | Anxiety symptoms (SCARED) | | |
| Model 1: adjusted for age and school attendance | *n* | IRR | [95%CI] | P>z | IRR | [95%CI] | *P>z* | *n* | IRR | [95%CI] | P>z | IRR | [95%CI] | P>z |
| Bodyweight Categories | 678 | | | | | | | 718 | | | | | | |
| Severe thinness (zBMI < −3 SD) | | 1.106 | 0.777,1.575 | .576 | 0.932 | 0.596,1.457 | .756 | | 1.064 | 0.869,1.303 | .548 | 1.036 | 0.815,1.317 | .771 |
| Thinness (−2SD > zBMI ≤ −3) | | 0.738 | 0.565,0.963 | **.025** | 1.090 | 0.845,1.406 | .507 | | 0.889 | 0.767,1.031 | .121 | 1.049 | 0.912,1.205 | .504 |
| Normal | | Ref | | | Ref | | | | Ref | | | Ref | | |
| Overweight/obese (zBMI > 2 SD) | | 0.496 | 0.25,0.982 | **.044** | 1.356 | 0.789,2.331 | .271 | | 0.673 | 0.467,0.968 | **.033** | 1.213 | 0.901,1.632 | .204 |
| Stunting + underweight | 678 | 0.937 | 0.675,1.30 | .697 | 0.948 | 0.789,1.140 | .570 | 718 | 1.182 | 0.877,1.594 | .273 | 1.005 | 0.852,1.185 | .955 |
| Household factors | | | | | | | | | | | | | | |
| Poor and poorest quartiles of the Wealth index | 678 | 1.004 | 0.807,1.248 | .957 | 1.087 | 0.961,1.230 | .183 | 718 | 1.055 | 0.871,1.279 | .583 | 1.037 | 0.934,1.151 | .499 |
| Financial support | 678 | 1.140 | 0.917,1.419 | .238 | 1.161 | 1.026,1.314 | **.018** | 718 | 1.113 | 0.922,1.345 | .266 | 1.047 | 0.945,1.160 | .383 |
| | | Depressive symptoms (SMFQ) | | | Anxiety symptoms (SCARED) | | | | Depressive symptoms (SMFQ) | | | Anxiety symptoms (SCARED) | | |
| Model 2: adjusted for age, school attendance, mother's occupation and the presence of any other micronutrient deficiency | | IRR | [95%CI] | P>z | IRR | [95%CI] | P>z | | IRR | [95%CI] | P>z | IRR | [95%CI] | P>z |
| Food diversity and insecurity | | | | | | | | | | | | | | |
| HDDS score | | 0.956 | 0.900,1.016 | .150 | 0.997 | 0.965,1.030 | .840 | | 0.956 | 0.905,1.010 | .105 | **0.964** | **0.936,0.993** | **.014** |
| Moderate/severe FIES | | **1.524** | 1.169,1.987 | **.002** | **1.328** | 1.141,1.545 | **<.001** | | 1.264 | 0.998,1.601 | .052 | **1.221** | **1.072,1.390** | **.003** |
| Folate | | | | | | | | | | | | | | |
| Possible deficiency | | **1.432** | 1.023,2.004 | **.036** | 1.031 | 0.858,1.240 | .743 | | **1.421** | 1.081,1.868 | **.012** | 1.127 | 0.976,1.303 | .104 |
| Deficiency | | **1.529** | 1.082,2.160 | **.016** | 1.200 | 0.994,1.450 | .058 | | **1.525** | 1.136,2.047 | **.005** | **1.219** | **1.041,1.427** | **.014** |
| Missing | | 1.311 | 0.868,1.981 | .198 | 1.174 | 0.936,1.474 | .166 | | **1.706** | 1.239,2.348 | **.001** | 1.153 | 0.970,1.371 | .105 |
| Bodyweight categories | | | | | | | | | | | | | | |
| Severe thinness (zBMI < −3 SD) | | 1.005 | 0.710,1.424 | .977 | 1.031 | 0.845,1.259 | .762 | | 0.808 | 0.522,1.251 | .340 | 0.968 | 0.766,1.224 | .789 |
| Thinness (−2SD > zBMI ≤ −3) | | 0.787 | 0.605,1.024 | .075 | 0.926 | 0.800,1.072 | .301 | | 1.187 | 0.925,1.522 | .177 | 1.092 | 0.953,1.251 | .204 |
| Normal | | Ref | | | Ref | | | | Ref | | | Ref | | |
| Overweight/obese (zBMI > 2 SD) | | 0.564 | 0.288,1.102 | .094 | 0.715 | 0.500,1.023 | .066 | | 1.160 | 0.682,1.971 | .585 | 1.143 | 0.854,1.529 | .368 |

(*Continued*)

**Table 2.** (*Continued*)

| Model 3: adjusted for age, school attendance, mother's occupation, mother's mental health and the presence of any other micronutrient deficiency | Depressive symptoms (SMFQ) | | | Anxiety symptoms (SCARED) | | | Depressive symptoms (SMFQ) | | | Anxiety symptoms (SCARED) | | |
|---|---|---|---|---|---|---|---|---|---|---|---|---|
| | IRR | [95%CI] | P>z | IRR | [95%CI] | P>z | IRR | [95%CI] | P>z | IRR | [95%CI] | P>z |
| Food diversity and insecurity | | | | | | | | | | | | |
| HDDS score | 0.981 | 0.928,1.037 | .493 | 1.008 | 0.978,1.038 | .617 | 1.001 | 0.952,1.054 | .935 | 0.982 | 0.955,1.009 | .193 |
| Moderate/severe FIES | 1.208 | 0.944,1.547 | .134 | 1.154 | 1.000,1.333 | .050 | 2.583 | 0.994,6.712 | .051 | 1.092 | 0.966,1.235 | .159 |
| Interaction FIES*mother's mental health | | | | | | | 0.980 | 0.961,1.000 | .042 | | | |
| Folate | | | | | | | | | | | | |
| Possible deficiency | **1.380** | 1.007,1.890 | **.045** | 1.024 | 0.862,1.217 | .785 | **1.310** | 1.017,1.688 | **.036** | 1.119 | 0.977,1.281 | .103 |
| Deficiency | **1.468** | 1.064,2.026 | **.019** | 1.166 | 0.978,1.391 | .087 | **1.315** | 1.000,1.730 | **.050** | **1.162** | **1.003,1.347** | **.046** |
| Missing | 0.978 | 0.662,1.445 | .910 | 1.086 | 0.878,1.344 | .448 | 1.655 | 1.235,2.219 | .001 | 1.143 | 0.973,1.342 | .105 |
| Bodyweight categories | | | | | | | | | | | | |
| Severe thinness (zBMI < −3 SD) | 1.133 | 0.823,1.561 | .444 | 1.050 | 0.872,1.266 | .606 | 0.742 | 0.493,1.116 | .152 | 0.966 | 0.776,1.202 | .754 |
| Thinness (−2SD > zBMI ≤ −3) | 0.862 | 0.674,1.102 | .236 | 0.990 | 0.863,1.136 | .890 | 1.013 | 0.803,1.278 | .914 | 1.010 | 0.889,1.148 | .875 |
| Normal | Ref | | | Ref | | | Ref | | | Ref | | |
| Overweight/obese (zBMI > 2 SD) | 0.679 | 0.358,1.287 | .235 | 0.797 | 0.569,1.116 | .186 | 1.292 | 0.792,2.107 | .305 | 1.218 | 0.928,1.598 | .155 |

*Notes*: other micronutrient deficiencies included vitamin A, Vitamin D, Folate, and Zinc.
Bold values indicate statistical significance.
Abbreviations: IRR: incident rate ratio; SMFQ: Short Mood and Feelings Questionnaire; SCARED: Screen for Child Anxiety-related Disorders; HDDS: Household Dietary Diversity Scale; BMI: Body Mass Index; FIES: Food Insecurity Experience Scale.

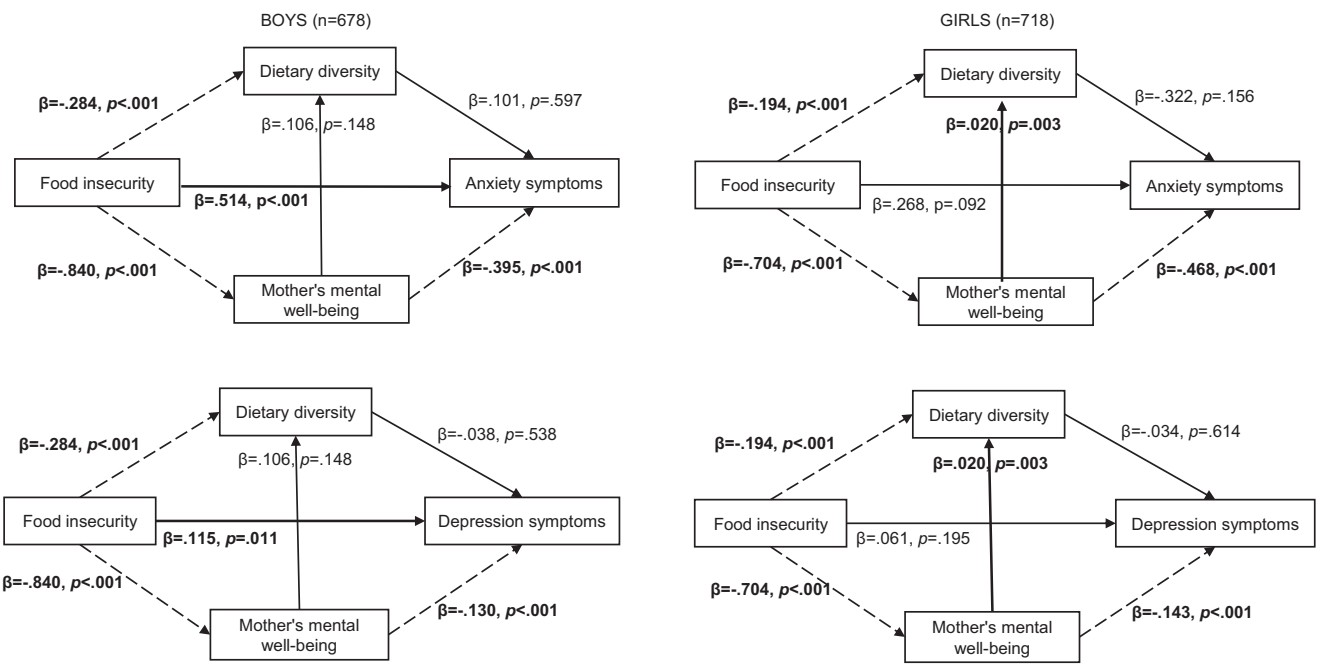

**Figure 2.** The role of maternal mental well-being in the relationship between food insecurity and mental health symptoms, by sex. *Note*: Dash arrows indicate a significant negative association; bold line arrows indicate a significant positive association, and black arrows indicate a non-significant association. Structural Equation Models are adjusted for age, school attendance, mother's occupation and BMI (categories).

studies showing similar connections between low dietary diversity and mental health issues in adults in LMIC (Jiang et al., 2018; Pengpid and Peltzer, 2024; Poorrezaeian et al., 2017) and in adolescents in Ethiopia (Tsehay et al., 2021). The strong correlation between household food insecurity experiences and adolescent mental health observed in this study further supports existing literature.

A particularly concerning finding is the high prevalence of multiple deficiencies (three or more) among both boys (29.9%) and girls (35.1%) and the extremely high prevalence of folate deficiency or possible deficiency among both boys (83.1%) and girls (78.4%), which is associated with elevated levels of depression and anxiety symptoms. This aligns with extensive research showing a clear link between low folate levels and increased risk of depression and anxiety in both adults (Gao et al., 2024; Liwinski and Lang, 2023) and adolescents (Campisi et al., 2020; Esnafoglu, 2023; Taj et al., 2024). Various mechanisms link nutrition and mental health through interconnected pathways. For instance, a diet deficient in folate – commonly found in leafy green vegetables, legumes, citrus fruits, whole grains and fortified foods – can impair brain function directly. Folate deficiency disrupts critical methylation processes in the central nervous system, leading to neurochemical imbalances that may contribute to mental health disorders (Black, 2008; Muscaritoli, 2021; Singh and Yadav, 2020). Additionally, household food insecurity experiences impact household dietary diversity, and the resulting stress and uncertainty can have a detrimental effect on mental health (Onyeaka et al., 2024; Pourmotabbed et al., 2020). The absence of differences in mental health symptoms across wealth gradients was likely attributable to the overall uniformity in poverty, as reflected by the proportion of households (57%) living below government-defined poverty thresholds and receiving financial support.

Despite the well-established association between excess body weight and increased mental health issues in many populations, our

findings suggest a potentially more complex relationship in rural LMIC settings among adolescents. Contrary to other literature, the current research found an association between obesity and low levels of depressive symptoms in unadjusted models in both boys and girls (Harikha and Kalalo, 2021; Moradi et al., 2021). However, these associations were attenuated by maternal occupation and micronutrient deficiencies (vitamin A, vitamin D, folate and zinc). One possible explanation is that our study was underpowered to detect an association between body weight and mental health outcomes, given the low prevalence of obesity/overweight in our sample (3% in boys and 3.1% in girls). Additional larger studies are needed to explore the relationship among body weight, nutrition and mental health in rural LMIC environments.

The finding from the current study underscores the critical role of nutrition in supporting adolescent mental health, highlighting the need to address nutritional deficiencies and ensure access to a diverse, nutrient-rich diet for this vulnerable population. Furthermore, the study provides evidence suggesting that maternal mental health may be associated with both food insecurity and adolescent mental health. This finding aligns with theoretical models like stress transmission (Goodman, 2020) and social–ecological frameworks (Batiari et al., 2022; Jang et al., 2020), which emphasize the broader context of parental well-being in shaping child outcomes (Duffy et al., 2023). While maternal mental health emerged as a key factor linking household food insecurity experiences to adolescent mental health in the current study, previous studies have shown that family dynamics, including paternal mental health, also play significant roles in adolescent well-being (Buehler, 2020; Luijten et al., 2021). These findings underscore the need for holistic, family-centred interventions that address the mental health of parents and children.

The relationship between maternal mental health and nutrition is potentially bidirectional. Mothers experiencing mental health disorders, such as depression or anxiety, may be less likely to

provide adequate emotional support or promote adaptive coping strategies, leading to adverse outcomes for adolescent mental health. Additionally, the link between maternal depression and poor household dietary diversity may reflect broader social determinants of health, with maternal mental health potentially serving as a proxy for overall family well-being and mental health.

This study's insights stem from its methodological strengths. First, it was conducted on a large rural population of adolescents in Pakistan, a demographic that has been understudied, allowing for the examination of various nutrition measures and maternal and household characteristics. Second, the study included adolescents attending and not attending school, thereby capturing information from non-school-going girls, a population often overlooked in adolescent research in LMIC. Third, trained psychologists administered all questionnaires, including the SMFQ and SCARED instruments, to minimize language and comprehension barriers. Despite these strengths, the study has limitations. As the school avoidance score was excluded from the total SCARED score, anxiety levels may have been underestimated. However, this is unlikely to have affected the associations presented, since all models controlled for school attendance. Although standardized psychometric tools were administered, they may not be culturally relevant, potentially affecting the accuracy of responses. Additionally, due to cultural sensitivities, adolescent questionnaires were administered in the presence of a parent or chaperone, which might have biased responses towards more socially or maternally acceptable answers. Furthermore, psychometric tools can only quantify symptomology. Thresholds of symptomology act as proxies for clinical diagnosis, which may have led to overestimation of the burden of depression and anxiety (Maxim et al., 2014). The HDDS and FIES are collected at the household level; therefore cannot provide any information on the intra-household food distribution or individual levels of food insecurity and dietary intake. Furthermore, household-level data may underestimate adolescent insecurity and dietary diversity due to discordant perceptions between parents and adolescents (Frank and Sato, 2024). In low- and middle-income countries (LMIC), food may be preferentially allocated to males in the household, potentially exacerbating sex disparities in food access (Ghatak et al., 2024; Rajan and Morgan, 2018). Additionally, adolescents have distinct nutritional requirements due to growth spurts, making them more vulnerable to food insecurity, even when household food security and dietary diversity appear adequate (Das et al., 2017). Nutrient deficiencies beyond those examined in the current study, such as omega-3 polyunsaturated fatty acids and vitamin $B_{12}$, may be important to consider for adolescent mental health, as these nutrients have been shown to play significant roles in supporting mental health in children and adolescents (Chang and Su, 2020; Tan et al., 2023). Additionally, the presence of gastrointestinal infections or other ongoing infections may influence dietary choices, contribute to nutritional deficiencies and present with non-specific depressive symptoms, which could confound associations observed between nutrition and mental health outcomes. The cross-sectional design of the study also prevents the establishment of temporal and causal relationships between the variables. Finally, the study was conducted in a low-income rural community, and the findings may not be generalizable to other populations.

The findings of this study have important implications for public health. They highlight the need for early interventions to address household food insecurity experiences and micronutrient deficiencies to improve adolescent mental health outcomes. In addition, investing in maternal mental health can have a positive impact on the mental health of children and adolescents. By targeting these factors, interventions can potentially disrupt the influence of stress, maternal stress and promote positive mental health outcomes for adolescents. Future research should employ longitudinal study designs, including adolescent dietary intake measures and examine different cultural and socioeconomic contexts.

## Conclusions

The current study contributes to a growing body of evidence highlighting the importance of addressing nutritional factors promoting household dietary diversity and maternal well-being to improve adolescent mental health, especially in vulnerable populations such as rural Pakistan. Future research is necessary to uncover the underlying mechanisms, explore the long-term implications and develop effective interventions.

**Open peer review.** To view the open peer review materials for this article, please visit http://doi.org/10.1017/gmh.2025.10006.

**Supplementary material.** The supplementary material for this article can be found at http://doi.org/10.1017/gmh.2025.10006.

**Data availability statement.** Data described in the manuscript, code book, and analytic code will be made available upon request to the corresponding author.

**Acknowledgements.** We thank the adolescents and their families who participated in the Nash-wo-Numa study and contributed to advancing adolescent research in rural Pakistan. We also thank the clinic staff for facilitating data collection. We acknowledge the valuable support provided by the entire team at the Aga Khan University.

**Author contribution.** Conceptualization: SCC, FP, YW, ZAB, PS; Methodology: SCC, FP, YWasan, ZAB, PS; Project administration: SCC, YW, SBS; Formal analysis and investigation: SCC, FP; Writing - original draft preparation: SCC, FP, YW; Writing - review and editing: SCC, FP, YW, SBS, DK, SM, PS, ZAB; Funding acquisition: SCC, ZAB; Supervision: ZAB, PS. All authors commented on previous versions of the manuscript. All authors read and approved the final manuscript.

**Financial support.** This work was supported by the Cundill Centre for Child and Youth Depression at the Centre for Addiction and Mental Health, Toronto, Canada.

**Competing interests.** The authors declared that they have no conflict of interest.

**Ethics statement.** This study received ethics approvals from the Aga Khan University, Karachi, Pakistan Ethics Review Committee and SickKids Hospital, Toronto, Canada Research Ethics Board, and the Centre for Addiction and Mental Health Research Ethics Committee. Informed written consent from parents or legal guardians and written assent from adolescents were obtained.

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
