## [Reviewer Report]

The article addresses an important topic i.e. association of food insecurity and nutrition on mental health of adolescents. The study was conducted in a rural district of Sindh, Pakistan on 9-15 year old girls and boys. Household level food insecurity and dietary diversity were assessed. Additionally, mother’s mental well-being and adolescents micronutrient levels were measured. Incidence Rate Ratios were estimated and study found association of mental well-being and household food insecurity with adolescent depression and anxiety through structural equation modelling.

Introduction:

1. Please add reference to lines 35 and 36 in the introduction.

2. Adolescents are typically grouped from age 10-19 years. Authors should provide the reference to categorize 9 year as adolescents.

Methods:

1. Participants were called to the field-office for interviews and that could have overwhelmed participants (both adolescents and mothers) and affected the mental health assessment. Please justify why participants were not interviewed at their homes.

2. Literature reported that adolescent food insecurity does not correlate with household food insecurity. Similarly, household dietary diversity may not reflect adolescent dietary diversity, especially girls‘ dietary diversity. In the authors’ opinion, how it had affected the study results.

3. Authors need to justify estimating IRR instead of OR or PR. IRR can be used for chronic diseases, however, it may be problematic in the study’s context. Depression and anxiety can lead to poor productivity, limited resource planning, and difficulty in managing crises pushing families towards food insecurity.

4. Please specify if married and unmarried both girls and boys were included. Married individuals can have unmeasured confounders of anxiety and depression. How the marital status was handled (if required)?

---

## [Reviewer Report]

I expect that adolescents will be part of your keyword as it is the target population.

Maternal health seems to be an afterthought that wasn’t included at first in the study and as such not much information was provided. i suggest including it into the bigger picture and justify its necessity.

Kindly check the overall language use for minor grammatical errors

---

## [Editor Report]

Dear Authors,

Your manuscript ‘ Impact of household food insecurity and nutrition on depression and anxiety symptoms among adolescents living in rural Pakistan’ has now been reviewed.

Please address the comments of both reviewers in addition to the following:

1) In section of procedure, you indicated that participants were interviewed from offices and not their home, natural environment, how did you deal with psychological distress that may have been present in your respondents so that it is not picked up as depressive or anxiety symptoms?

2) Under Mental Health Measures, the authors state that the scores for SMFQ is 0 to 26 and also 12 to 60. This is confusing, please clarify. Also, SMFQ has different cut off points for different studies. In the original study which is included in your references, a cut off of 8 is considered significant at a Sensitivity of 60% and specificity of 85% for major depression(Source is 

Angold A, Costello EJ, Messer SC. “Development of a short questionnaire for use in epidemiological studies of 

depression in children and adolescents.” International Journal of Methods in Psychiatric Research (1995), 

5:237-249). What cut off point was used in this study?

3) For measures of anxiety, the questionnaire SCARED was used. The authors state that the subscale of school avoidance was left out as only about 52% girls and 75% boys attended school. what was the reason for those not attending school? Is it possible that it could have been school avoidance?

Secondly what could have been the effects on internal validity of the questionnaire having left or the subscale of school avoidance?

What was the cuff of the subscales?

---

## [Editor Report]

Dear Author,

Your revised manuscript :’Impact of household food insecurity and nutrition on depression and anxiety symptoms among adolescents living in rural Pakistan', has been reviewed.